# Sound and Complete Verification of Polynomial Networks

**Elias Abad Rocamora**[*]
LIONS, EPFL
Lausanne, Switzerland
abad.elias00@gmail.com

**Mehmet Fatih Sahin**
LIONS, EPFL
Lausanne, Switzerland
mehmet.sahin@epfl.ch

**Fanghui Liu**
LIONS, EPFL
Lausanne, Switzerland
fanghui.liu@epfl.ch

**Grigorios G Chrysos**
LIONS, EPFL
Lausanne, Switzerland
grigorios.chrysos@epfl.ch

**Volkan Cevher**
LIONS, EPFL
Lausanne, Switzerland
volkan.cevher@epfl.ch

## Abstract

Polynomial Networks (PNs) have demonstrated promising performance on face and image recognition recently. However, robustness of PNs is unclear and thus obtaining certificates becomes imperative for enabling their adoption in real-world applications. Existing verification algorithms on ReLU neural networks (NNs) based on classical branch and bound (BaB) techniques cannot be trivially applied to PN verification. In this work, we devise a new bounding method, equipped with BaB for global convergence guarantees, called Verification of Polynomial Networks or VPN for short. One key insight is that we obtain much tighter bounds than the interval bound propagation (IBP) and DeepT-Fast [Bonaert et al., 2021] baselines. This enables sound and complete PN verification with empirical validation on MNIST, CIFAR10 and STL10 datasets. We believe our method has its own interest to NN verification. The source code is publicly available at https://github.com/megaelius/PNVerification.

## 1 Introduction

Polynomial Networks (PNs) have demonstrated promising performance across image recognition and generation [Chrysos et al., 2021, Chrysos and Panagakis, 2020] being state-of-the-art on large-scale face recognition[2].Unlike the conventional Neural Networks (NNs), where non-linearitiy is introduced with the use of activation functions [LeCun et al., 2015], PNs are able to learn non-linear mappings without the need of activation functions by exploiting multiplicative interactions (Hadamard products). Recent works have uncovered interesting properties of PNs in terms of model expressivity [Fan et al., 2021] and spectral bias [Choraria et al., 2022]. However, one critical issue before considering PNs for real-world applications is their robustness.

Neural networks are prone to small (often imperceptible to the human eye), but malicious perturbations in the input data points [Szegedy et al., 2014, Goodfellow et al., 2015]. Those perturbations can have a detrimental effect on image recognition systems, e.g., as illustrated in face recognition [Goswami et al., 2019, Zhong and Deng, 2019, Dong et al., 2019, Li et al., 2020]. Guarding against such attacks has so far proven futile [Shafahi et al., 2019, Dou et al., 2018]. Instead, a flurry of research

---

[*]Work developed during an exchange coming from Universitat Politècnica de Catalunya (UPC), Spain. Currently at Universidad Carlos III de Madrid (UC3M).

[2]https://paperswithcode.com/sota/face-verification-on-megaface

has been published on certifying robustness of NNs against this performance degradation [Katz et al., 2017, Ehlers, 2017, Tjeng et al., 2019, Bunel et al., 2020b, Wang et al., 2021, Ferrari et al., 2022]. However, most of the verification algorithms for NNs are developed for the ReLU activation function by exploiting its piecewise linearity property and might not trivially extend to other nonlinear activation functions [Wang et al., 2021]. Indeed, Zhu et al. [2022] illustrate that guarding PNs against adversarial attacks is challenging. Therefore, we pose the following question:

*Can we obtain certifiable performance for PNs against adversarial attacks?*

In this work, we answer affirmatively and provide a method for the verification of PNs. Concretely, we take advantage of the twice-differentiable nature of PNs to build a lower bounding method based on $\alpha$-convexification [Adjiman and Floudas, 1996], which is integrated into a Branch and Bound (BaB) algorithm [Land and Doig, 1960] to guarantee completeness of our verification method. In order to use $\alpha$-convexification, a lower bound $\alpha$ of the minimum eigenvalue of the Hessian matrix over the possible perturbation set is needed. We use interval bound propagation together with the theoretical properties of the lower bounding Hessian matrix [Adjiman et al., 1998], in order to develop an algorithm to efficiently compute $\alpha$.

Our *contributions* can be summarized as follows: $(i)$ We propose the first algorithm for the verification of PNs. $(ii)$ We thoroughly analyze the performance of our method by comparing it with a black-box solver, with an interval bound propagation (IBP) BaB algorithm and with the zonotope-based abstraction method DeepT-Fast [Bonaert et al., 2021]. $(iii)$ We empirically show that using $\alpha$-convexification for lower bounding provides tighter bounds than IBP and DeepT-Fast for PN verification. To encourage the community to improve the verification of PNs, we make our code publicly available in `https://github.com/megaelius/PNVerification`. The proposed approach can practically verify PNs, while it could also theoretically be applied for sound and complete verification of any twice-differentiable network.

**Notation:** We use the shorthand $[n] := \{1, 2, \ldots, n\}$ for a positive integer $n$. We use bold capital (lowercase) letters, e.g., $\boldsymbol{X}$ $(\boldsymbol{x})$ for representing matrices (vectors). The $j^{\text{th}}$ column of a matrix $\boldsymbol{X}$ is given by $\boldsymbol{x}_{:j}$. The element in the $i^{\text{th}}$ row and $j^{\text{th}}$ column is given by $x_{ij}$, similarly, the $i^{\text{th}}$ element of a vector $\boldsymbol{x}$ is given by $x_i$. The element-wise (Hadamard) product, symbolized with $*$, of two matrices (or vectors) in $\mathbb{R}^{d_1 \times d_2}$ (or $\mathbb{R}^d$) gives another matrix (or vector) in $\mathbb{R}^{d_1 \times d_2}$ (or $\mathbb{R}^d$). The $\ell_\infty$ norm of a vector $\boldsymbol{x} \in \mathbb{R}^d$ is given by: $||\boldsymbol{x}||_\infty = \max_{i \in [d]} |x_i|$. Lastly, the operators $\mathcal{L}$ and $\mathcal{U}$ give the lower and upper bounds of a scalar, vector or matrix function by IBP, see Section 3.1.

**Roadmap:** We provide the necessary background by introducing the PN architecture and formalizing the Robustness Verification problem in Section 2. Section 3 provides a *sound* and *complete* method called VPN to tackle PN verification problem. Section 4 is devoted to experimental validation. Additional experiments, details and proofs are deferred to the appendix.

## 2 Background

We give an overview of the PN architecture in Section 2.1 and the robustness verification problem in Section 2.2.

### 2.1 Polynomial Networks (PNs)

Polynomial Networks (PNs) are inspired by the fact that any smooth function can be approximated via a polynomial expansion [Stone, 1948]. However, the number of parameters increases exponentially with the polynomial degree, which makes it intractable to use high degree polynomials for high-dimensional data problems such as image classification where the input can be in the order of $10^5$ [Deng et al., 2009]. Chrysos et al. [2021] introduce a joint factorization of polynomial coefficients in a low-rank manner, reducing the number of parameters to linear with the polynomial degree and allowing the expression as a neural network (NN). We briefly recap one fundamental factorization below.

Let $N$ be the polynomial degree, $\boldsymbol{z} \in \mathbb{R}^d$ be the input vector, $d$, $k$ and $o$ be the input, hidden and output sizes, respectively. The recursive equation of PNs can be expressed as:

$$\boldsymbol{x}^{(n)} = (\boldsymbol{W}_{[n]}^\top \boldsymbol{z}) * \boldsymbol{x}^{(n-1)} + \boldsymbol{x}^{(n-1)}, \forall\, n \in [N], \tag{1}$$

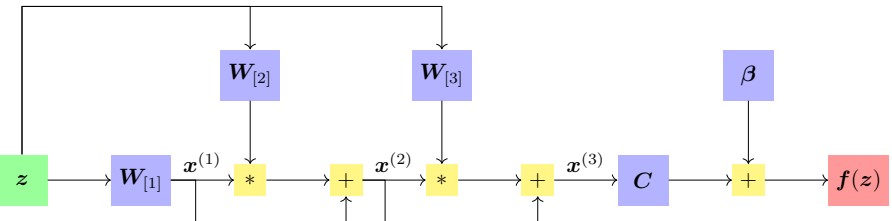

Figure 1: Third degree PN architecture. Blue boxes depict learnable parameters, yellow depict mathematical operations, the green and red boxes are the input and the output respectively. Note that no activation functions are involved, only element-wise (Hadamard) products $*$ and additions $+$. This figure represents the recursive formula of Eq. (1).

where $\boldsymbol{x}^{(1)} = \boldsymbol{W}_{[1]}^{\top}\boldsymbol{z}$, $\boldsymbol{f}(\boldsymbol{z}) = \boldsymbol{C}\boldsymbol{x}^{(N)} + \boldsymbol{\beta}$ and $*$ denotes the Hadamard product. $\boldsymbol{W}_{[n]} \in \mathbb{R}^{d \times k}$ and $\boldsymbol{C} \in \mathbb{R}^{o \times k}$ are weight matrices, $\boldsymbol{\beta} \in \mathbb{R}^{o}$ is a bias vector. A graphical representation of a third degree PN architecture corresponding to Eq. (1) can be found in Fig. 1. Further details on the factorization (as well as other factorizations) are deferred to the Appendix C.1 (Appendix C.2).

## 2.2 Robustness Verification

Robustness verification [Bastani et al., 2016, Liu et al., 2021] consists of verifying that a property regarding the input and output of a NN is satisfied, e.g. checking whether or not a small perturbation in the input will produce a change in the network output that makes it classify the input into another class. Let $f : [0, 1]^d \rightarrow \mathbb{R}^o$ be a function, e.g., a NN or a PN, that classifies the input $\boldsymbol{z}$ into a class $c$, such that $c = \arg\max \boldsymbol{f}(\boldsymbol{z})$. Our target is to verify that for any input satisfying a set of constraints $C_{\text{in}}$, the output of the network will satisfy a set of output constraints $C_{\text{out}}$. Mathematically,

$$\boldsymbol{z} \in C_{\text{in}} \implies \boldsymbol{f}(\boldsymbol{z}) \in C_{\text{out}}. \tag{2}$$

In this work we focus on *adversarial robustness* [Szegedy et al., 2014, Carlini and Wagner, 2017] in classification. Given an observation $\boldsymbol{z}_0$, let $t = \arg\max \boldsymbol{f}(\boldsymbol{z}_0)$ be the correct class, our goal is to check whether every input in a neighbourhood of $\boldsymbol{z}_0$, is classified as $t$. In this work, we focus on adversarial attacks restricted to neighbourhoods defined in terms of $\ell_\infty$ norm, which is a popular norm-bounded attack in the verification community [Liu et al., 2021]. Then, the constraint sets become:

$$\begin{aligned} C_{\text{in}} &= \{\boldsymbol{z} : ||\boldsymbol{z} - \boldsymbol{z}_0||_\infty \leq \epsilon, z_i \in [0,1], \forall i \in [d]\} \\ &= \{\boldsymbol{z} : \max\{0, z_{0_i} - \epsilon\} \leq z_i \leq \min\{1, z_{0_i} + \epsilon\}, \forall i \in [d]\} \\ C_{\text{out}} &= \{\boldsymbol{y} : y_t > y_j, \forall j \neq t\}. \end{aligned} \tag{3}$$

In other words, we need an algorithm that given a function $f$, an input $\boldsymbol{z}_0$ and an adversarial budget $\epsilon$, checks whether Eq. (2) is satisfied. In the case of ReLU NNs, this has been proven to be an NP-complete problem [Katz et al., 2017]. This can be reformulated as a constrained optimization problem. For every adversarial class $\gamma \neq t = \arg\max \boldsymbol{f}(\boldsymbol{z}_0)$, we can solve:

$$\min_{\boldsymbol{z}} \quad g(\boldsymbol{z}) = f(\boldsymbol{z})_t - f(\boldsymbol{z})_\gamma \quad \text{s.t.} \quad \boldsymbol{z} \in \mathcal{C}_{\text{in}}. \tag{4}$$

If the solution $\boldsymbol{z}^*$ with $v^* = f(\boldsymbol{z}^*)_t - f(\boldsymbol{z}^*)_\gamma \leq f(\boldsymbol{z})_t - f(\boldsymbol{z})_\gamma, \forall \boldsymbol{z} \in \mathcal{C}_{\text{in}}$ satisfies $v^* > 0$ then robustness is verified for the adversarial class $\gamma$.

There are two main properties that a verification algorithm admits: *soundness* and *completeness*. An algorithm is *sound* (*complete*) if every time it verifies (falsifies) a property, it is guaranteed to be the correct answer. In practice, when an algorithm is guaranteed to provide the exact global minima of Eq. (4), i.e., $v^*$, it is said to be *sound* and *complete* (usually referred in the literature as simply *complete* [Ferrari et al., 2022]), whereas if a lower bound of it is provided $\hat{v}^* \leq v^*$, the algorithm is *sound* but not *complete*. In our work, we will not consider just *complete* verification, which simply aims at looking for adversarial examples, e.g., Madry et al. [2018]. For a deeper discussion on *soundness* and *completeness*, we refer to Liu et al. [2021].

# 3 Method

Our method, called VPN, can be categorized in the the Branch and Bound (BaB) framework [Land and Doig, 1960], a well known approach to global optimization [Horst and Tuy, 1996] and NN verification [Bunel et al., 2020b]. This kind of algorithms ensures finding a global minima of the problem in Eq. (4) by recursively splitting the original feasible set into smaller sets (branching) where upper and lower bounds of the global minima are computed (bounding). This mechanism can be used to discard subsets where the global minima cannot be achieved (its lower bound is greater than the upper bound of another subset).

Our method is based on a variant of BaB algorithm, i.e., $\alpha$-BaB [Adjiman et al., 1998], which is characterized for using $\alpha$-convexification [Adjiman and Floudas, 1996] for computing a lower bound of the global minima of each subset. To be specific, $\alpha$-convexification aims to obtain a convex lower bounding function of any twice-differentiable function $f : \mathbb{R}^d \to \mathbb{R}$. In Adjiman and Floudas [1996], they propose two methods:

- **Uniform diagonal shift (single $\alpha$)**

$$g_\alpha(\boldsymbol{z}; \alpha, \boldsymbol{l}, \boldsymbol{u}) = g(\boldsymbol{z}) + \alpha \sum_{i=1}^{d} (z_i - l_i)(z_i - u_i) \,, \tag{5}$$

  is its $\alpha$-convexified version, note that $z_i$ is the $i^{\text{th}}$ element of vector $\boldsymbol{z}$. Let $\boldsymbol{H}_g(\boldsymbol{z}) = \nabla^2_{\boldsymbol{zz}} g(\boldsymbol{z})$ be the Hessian matrix of $g$, $g_\alpha$ is convex in $\boldsymbol{z} \in [\boldsymbol{l}, \boldsymbol{u}]$ for $\alpha \geq \max\{0, -\frac{1}{2} \min\{\lambda_{\min}(\boldsymbol{H}_g(\boldsymbol{z})) : \boldsymbol{z} \in [\boldsymbol{l}, \boldsymbol{u}]\}\}$, where $\lambda_{\min}$ is the minimum eigenvalue. Moreover, it holds that $g_\alpha(\boldsymbol{z}; \alpha, \boldsymbol{l}, \boldsymbol{u}) \leq g(\boldsymbol{z}), \forall \boldsymbol{z} \in [\boldsymbol{l}, \boldsymbol{u}]$.

- **Non-uniform diagonal shift (multiple $\alpha$'s)**

$$g_{\boldsymbol{\alpha}}(\boldsymbol{z}; \boldsymbol{\alpha}, \boldsymbol{l}, \boldsymbol{u}) = g(\boldsymbol{z}) + \sum_{i=1}^{d} \alpha_i (z_i - l_i)(z_i - u_i) \,, \tag{6}$$

  is its $\alpha$-convexified version. In Adjiman et al. [1998], they show that for any vector $\boldsymbol{d} > \boldsymbol{0}$, setting

$$\alpha_i \geq \max \left\{ 0, -\frac{1}{2} \left( \mathcal{L}(h_g(\boldsymbol{z})_{ii}) - \sum_{j \neq i} \max\{|\mathcal{L}(h_g(\boldsymbol{z})_{ij})|, |\mathcal{U}(h_g(\boldsymbol{z})_{ij})|\} \frac{d_j}{d_i} \right) \right\} \tag{7}$$

  makes $g_{\boldsymbol{\alpha}}$ convex in $\boldsymbol{z} \in [\boldsymbol{l}, \boldsymbol{u}]$. The choice of the vector $\boldsymbol{d}$ is arbitrary, but affects the final result. For example, taking $\boldsymbol{d} = \boldsymbol{u} - \boldsymbol{l}$ yields better results than $\boldsymbol{d} = \boldsymbol{1}$ in Adjiman et al. [1998]. We need to remark that, though Eq. (5) is a special case of Eq. (6), the lower bound of $\alpha$ in Eq. (5) via minimum eigenvalue of the lower bounding Hessian matrix cannot be regarded as a special case of Eq. (7).

To make PN verification feasible via $\alpha$-convexification, we need to study IBP for PNs and design an efficient estimate on the $\alpha$ ($\boldsymbol{\alpha}$ in the case of Non-uniform diagonal shift) parameter, which are our main technical contributions in the algorithmic aspect. In our case, every feasible set, starting with the input set $\mathcal{C}_{\text{in}}$ (Eq. (3)), is split by taking the widest variable interval and dividing it in two by the middle point. This is a rather simple, but theoretically powerful strategy, see Lemma 8 in Appendix F. Then, the upper bound of each subproblem is given by applying standard Projected Gradient Descent (PGD) [Kelley, 1999] over the original objective function. This is a common approach to find adversarial examples [Madry et al., 2018], but as the objective is non-convex, it is not sufficient for sound and complete verification. The lower bound is given by applying PGD over the $\alpha$-convexified objective $g_\alpha$ ($g_{\boldsymbol{\alpha}}$), as it is convex, PGD converges to the global minima and a lower bound of the original objective. The $\alpha$ ($\boldsymbol{\alpha}$) parameter is computed only once per verification problem. Further details on the algorithm and the proof of convergence of Eq. (4) exist in Appendix D. A schematic of our method is available in Fig. 2.

In Sections 3.1 to 3.3, we detail our method under the uniform diagonal shift case to compute a lower bound on the minimum eigenvalue of the Hessian matrix into three main components: interval propagation, lower bounding Hessians, and fast estimation on such lower bounding via power method. To conclude the description of our method, in Section 3.4 we describe the $\boldsymbol{\alpha}$ estimation for the non-uniform diagonal shift case.

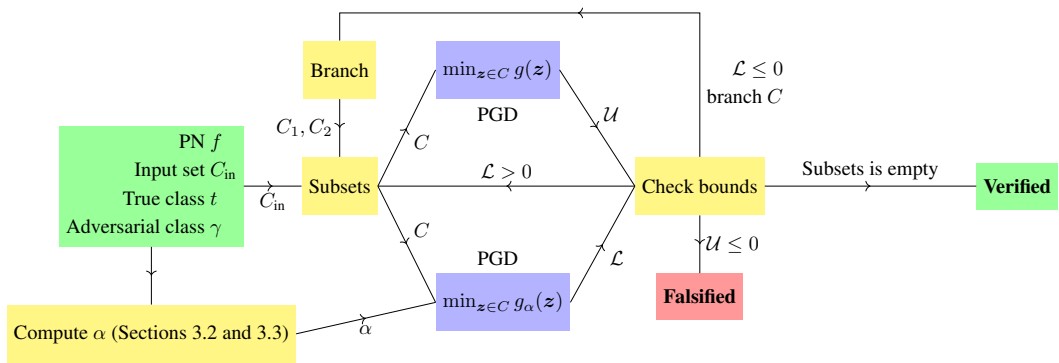

Figure 2: Overview of our branch and bound verification algorithm. Given a trained PN $f$, an input set $C_{\text{in}}$, the true class $t$ and an adversarial class $\gamma$, we check if an adversarial example exists (**Falsified**) or not (**Verified**). Note that the branching of a subset $C$ provides two smaller subsets $C_1$ and $C_2$. Also note that when $\mathcal{L} > 0$, no subset is added to subsets.

## 3.1 Interval Bound Propagation through a PN

Interval bound propagation (IBP) is a key ingredient of our verification algorithm. Suppose we have an input set defined by an $\ell_\infty$-norm ball like in Eq. (3). This set can be represented as a vector of intervals $[\boldsymbol{l}, \boldsymbol{u}] = ([l_1, u_1]^\top, [l_2, u_2]^\top, \cdots, [l_d, u_d]^\top) \in \mathbb{R}^{d \times 2}$, where $[l_i, u_i]$ are the lower and upper bound for the $i^{\text{th}}$ coordinate. Let $\mathcal{L}$ and $\mathcal{U}$ be the lower and upper bound IBP operators. Given this input set, we would like to obtain bounds on the output of the network $(f(\boldsymbol{z})_i)$, the gradient $(\nabla_{\boldsymbol{z}} f(\boldsymbol{z})_i)$, and the Hessian $(\nabla^2_{\boldsymbol{zz}} f(\boldsymbol{z})_i)$ for any $\boldsymbol{z} \in [\boldsymbol{l}, \boldsymbol{u}]$. The operators $\mathcal{L}(g(\boldsymbol{z}))$ and $\mathcal{U}(g(\boldsymbol{z}))$ of any function $g : \mathbb{R}^d \to \mathbb{R}$ satisfy:

$$\mathcal{L}(g(\boldsymbol{z})) \le g(\boldsymbol{z}), \quad \mathcal{U}(g(\boldsymbol{z})) \ge g(\boldsymbol{z}), \forall \boldsymbol{z} \in [\boldsymbol{l}, \boldsymbol{u}]. \tag{8}$$

We will define these upper and lower bound operators in terms of the operations present in a PN. Using interval propagation [Moore et al., 2009], denote the positive part $w_i^+ = \max\{0, w_i\}$ and the negative part $w_i^- = \min\{0, w_i\}$, $h_i(\boldsymbol{z})$ as a real-valued function of $\boldsymbol{z}$, $i \in [d]$, we can define:

$$\textbf{Identity} \begin{cases} \mathcal{L}(z_i) = l_i \\ \mathcal{U}(z_i) = u_i \end{cases}$$

$$\textbf{linear mapping} \begin{cases} \mathcal{L}(\sum_i w_i h_i(\boldsymbol{z})) = \sum_i w_i^+ \, \mathcal{L}(h_i(\boldsymbol{z})) + w_i^- \, \mathcal{U}(h_i(\boldsymbol{z})) \\ \mathcal{U}(\sum_i w_i h_i(\boldsymbol{z})) = \sum_i w_i^- \, \mathcal{L}(h_i(\boldsymbol{z})) + w_i^+ \, \mathcal{U}(h_i(\boldsymbol{z})) \end{cases}$$

$$\textbf{multiplication} \begin{cases} S = \begin{cases} \mathcal{L}(h_1(\boldsymbol{z})) \, \mathcal{L}(h_2(\boldsymbol{z})), \\ \mathcal{L}(h_1(\boldsymbol{z})) \, \mathcal{U}(h_2(\boldsymbol{z})), \\ \mathcal{U}(h_1(\boldsymbol{z})) \, \mathcal{L}(h_2(\boldsymbol{z})), \\ \mathcal{U}(h_1(\boldsymbol{z})) \, \mathcal{U}(h_2(\boldsymbol{z})) \end{cases}, |S| = 4 \\ \mathcal{L}(h_1(\boldsymbol{z}) h_2(\boldsymbol{z})) = \min S, \\ \mathcal{U}(h_1(\boldsymbol{z}) h_2(\boldsymbol{z})) = \max S, \end{cases}$$

$$\tag{9}$$

where $|\cdot|$ is the set cardinality. Note that the set $S$ is equivalent to:

$$S = \left\{ ab \middle| \, \forall a \in \{\mathcal{L}(h_1(\boldsymbol{z})), \mathcal{U}(h_1(\boldsymbol{z}))\}, \forall b \in \{\mathcal{L}(h_2(\boldsymbol{z})), \mathcal{U}(h_2(\boldsymbol{z}))\} \right\}.$$

With these basic operations, one can define bounds on any intermediate output, gradient or Hessian of a PN. For instance, the lower bound on the recursive formula from Eq. (1) can be expressed as:

$$\mathcal{L}(x_i^{(n)}) = \mathcal{L}((\boldsymbol{w}_{[n]:i}^\top \boldsymbol{z}) x_i^{(n-1)} + x_i^{(n-1)}) = \mathcal{L}((\boldsymbol{w}_{[n]:i}^\top \boldsymbol{z} + 1) x_i^{(n-1)}), \quad \forall i \in [k], n \in \{2, \ldots, N\}, \tag{10}$$

which only consists on a linear mapping and a multiplication of intervals. We extend the upper and lower bound ($\mathcal{L}(\cdot)$ and $\mathcal{U}(\cdot)$) operators to vectors and matrices in an entry-wise style:

$$
\mathcal{L}(\boldsymbol{g}(\boldsymbol{z})) = \begin{bmatrix} \mathcal{L}(g(\boldsymbol{z})_1) \\ \mathcal{L}(g(\boldsymbol{z})_2) \\ \vdots \\ \mathcal{L}(g(\boldsymbol{z})_m) \end{bmatrix} \in \mathbb{R}^m, \quad \mathcal{L}(\boldsymbol{G}(\boldsymbol{z})) = \begin{bmatrix} \mathcal{L}(g(\boldsymbol{z})_{11}) & \cdots & \mathcal{L}(g(\boldsymbol{z})_{1m}) \\ \vdots & \ddots & \\ \mathcal{L}(g(\boldsymbol{z})_{m1}) & & \mathcal{L}(g(\boldsymbol{z})_{mm}) \end{bmatrix} \in \mathbb{R}^{m \times m}.
\tag{11}
$$

Note that Eq. (11) is not limited to squared matrices and can hold for arbitrary matrix dimensions. One can directly use IBP to obtain bounds on the verification objective from Eq. (4) with a single forward pass of the bounds through the network and obtaining $\mathcal{L}(g(\boldsymbol{z})) = \mathcal{L}(f(\boldsymbol{z})_t) - \mathcal{U}(f(\boldsymbol{z})_\gamma)$. In fact IBP is a common practice in NN verification to obtain fast bounds [Wang et al., 2018a].

### 3.2 Lower bound of the minimum eigenvalue of the Hessian

Here we describe our method to compute a lower bound on the minimum eigenvalue of the Hessian matrix in the feasible set. Before deriving the lower bound, we need the first and second order partial derivatives of PNs.

Let $g(\boldsymbol{z}) = f(\boldsymbol{z})_t - f(\boldsymbol{z})_a$ be the objective function for $t = \arg\max \boldsymbol{f}(\boldsymbol{z}_0)$ and any $a \neq t$. In order to compute the parameter $\alpha$ for performing $\alpha$-convexification, we need to know the structure of our objective function. In this section we compute the first and second order partial derivatives of the PN. The gradient and Hessian matrices of the objective function (see Eq. (4)) are given by:

$$
\nabla_{\boldsymbol{z}} g(\boldsymbol{z}) = \sum_{i=1}^{k} (c_{ti} - c_{\gamma i}) \nabla_{\boldsymbol{z}} x_i^{(N)}, \quad \boldsymbol{H}_g(\boldsymbol{z}) = \sum_{i=1}^{k} (c_{ti} - c_{\gamma i}) \nabla_{\boldsymbol{z}\boldsymbol{z}}^2 x_i^{(N)}.
\tag{12}
$$

We now define the gradients $\nabla_{\boldsymbol{z}} x_i^{(n)}$ and Hessians $\nabla_{\boldsymbol{z}\boldsymbol{z}}^2 x_i^{(n)}$ of Eq. (1) in a recursive way:

$$
\nabla_{\boldsymbol{z}} x_i^{(n)} = \boldsymbol{w}_{[n]:i} \cdot x_i^{(n-1)} + (\boldsymbol{w}_{[n]:i}^\top \boldsymbol{z} + 1) \cdot \nabla_{\boldsymbol{z}} x_i^{(n-1)}
\tag{13}
$$

$$
\nabla_{\boldsymbol{z}\boldsymbol{z}}^2 x_i^{(n)} = \nabla_{\boldsymbol{z}} x_i^{(n-1)} \boldsymbol{w}_{[n]:i}^\top + \{\nabla_{\boldsymbol{z}} x_i^{(n-1)} \boldsymbol{w}_{[n]:i}^\top\}^\top + (\boldsymbol{w}_{[n]:i}^\top \boldsymbol{z} + 1) \nabla_{\boldsymbol{z}\boldsymbol{z}}^2 x_i^{(n-1)},
\tag{14}
$$

with $\nabla_{\boldsymbol{z}} x_i^{(1)} = \boldsymbol{w}_{[1]:i}$ and $\nabla_{\boldsymbol{z}\boldsymbol{z}}^2 x_i^{(1)} = \boldsymbol{0}_{d \times d}$ being an all-zero matrix. In the next, we are ready to compute a lower bound on the minimum eigenvalue of the Hessian matrix in the feasible set.

Firstly, for any $\boldsymbol{z} \in [\boldsymbol{l}, \boldsymbol{u}]$ and any polynomial degree $N$, we can express the set of possible Hessians $\mathcal{H} = \{\boldsymbol{H}_g(\boldsymbol{z}) : \boldsymbol{z} \in [\boldsymbol{l}, \boldsymbol{u}]\}$ as an interval matrix. An interval matrix is a tensor $[\boldsymbol{M}] \in \mathbb{R}^{d \times d \times 2}$ where every position $[m]_{ij} = [\mathcal{L}(m_{ij}), \mathcal{U}(m_{ij})]$ is an interval. Therefore, if $\boldsymbol{H}_g(\boldsymbol{z})$ is bounded for $\boldsymbol{z} \in [\boldsymbol{l}, \boldsymbol{u}]$, then we can represent $\mathcal{H} = \{\boldsymbol{H}_g(\boldsymbol{z}) : \boldsymbol{H}_g(\boldsymbol{z}) \in [\boldsymbol{M}]\} = \{\boldsymbol{H}_g(\boldsymbol{z}) : \mathcal{L}(m_{ij}) \leq \boldsymbol{H}_g(\boldsymbol{z})_{ij} \leq \mathcal{U}(m_{ij}), \forall i, j \in [d]\}$.

Let $\mathcal{L}(\boldsymbol{M})$ and $\mathcal{U}(\boldsymbol{M})$ be the element-wise lower and upper bounds of a Hessian matrix, the lower bounding Hessian is defined as follows:

$$
\boldsymbol{L_H} = \frac{\mathcal{L}(\boldsymbol{M}) + \mathcal{U}(\boldsymbol{M})}{2} + \operatorname{diag}\left(\frac{\mathcal{L}(\boldsymbol{M})\boldsymbol{1} - \mathcal{U}(\boldsymbol{M})\boldsymbol{1}}{2}\right),
\tag{15}
$$

where $\boldsymbol{1}$ is an all-one vector and $\operatorname{diag}(\boldsymbol{v})$ is a diagonal matrix with the vector $\boldsymbol{v}$ in the diagonal. Described in Adjiman et al. [1998], this matrix satisfies that $\lambda_{\min}(\boldsymbol{L_H}) \leq \lambda_{\min}(\boldsymbol{H}_g(\boldsymbol{z})), \forall \boldsymbol{H}_g(\boldsymbol{z}) \in \mathcal{H}, \boldsymbol{z} \in [\boldsymbol{l}, \boldsymbol{u}]$.

Then, we can obtain the spectral radius $\rho(\boldsymbol{L_H})$ with a power method [Mises and Pollaczek-Geiringer, 1929]. As the spectral radius satisfies $\rho(\boldsymbol{L_H}) \geq |\lambda_i(\boldsymbol{L_H})|, \forall i \in [d]$, the following inequality holds:

$$
-\rho(\boldsymbol{L_H}) \leq \lambda_{\min}(\boldsymbol{L_H}) \leq \lambda_{\min}(\boldsymbol{H}_g(\boldsymbol{z})), \ \forall \boldsymbol{H}_g(\boldsymbol{z}) \in \mathcal{H}, \ \boldsymbol{z} \in [\boldsymbol{l}, \boldsymbol{u}],
\tag{16}
$$

allowing us to use $\alpha = \frac{\rho(\boldsymbol{L_H})}{2} \geq \max\{0, -\frac{1}{2}\min\{\lambda_{\min}(\boldsymbol{H}_f(\boldsymbol{z})) : \boldsymbol{z} \in [\boldsymbol{l}, \boldsymbol{u}]\}\}$.

### 3.3 Efficient power method for spectral radius computation of the lower bounding Hessian

By using interval propagation, one can easily compute sound lower and upper bounds on each position of the Hessian matrix, compute the lower bounding Hessian and perform a power method with it to

obtain the spectral radius $\rho$. However, this method would not scale well to high dimensional scenarios. For instance, in the STL10 dataset [Coates et al., 2011] (with input dimension $d = 96{\cdot}96{\cdot}3 = 27,648$) in each color image, our Hessian matrix would require an order of $O(d^2) = O(10^9)$ real numbers to be stored. This makes it intractable to perform a power method over such an humongous matrix, or even to compute the lower bounding Hessian. Alternatively, we take advantage of the possibility of expressing the $\boldsymbol{L_H}$ matrix as a sum of rank-1 matrices, to enable performing a power method over it.

**Standard power method for spectral radius computation**

Given any squared and real valued matrix $\boldsymbol{M} \in \mathbb{R}^{d \times d}$ and an initial vector $\boldsymbol{v}_0 \in \mathbb{R}^d$ that is not an eigenvector of $\boldsymbol{M}$, the sequence:

$$\boldsymbol{v}_n = \frac{\boldsymbol{M}(\boldsymbol{M}\boldsymbol{v}_{n-1})}{||\boldsymbol{M}(\boldsymbol{M}\boldsymbol{v}_{n-1})||_2}\,, \tag{17}$$

converges to the eigenvector with the largest eigenvalue in absolute value, i.e. the eigenvector where the spectral radius is attained, being the spectral radius $\rho(\boldsymbol{M}) = \sqrt{||\boldsymbol{M}(\boldsymbol{M}\boldsymbol{v}_{n-1})||_2}$ [Mises and Pollaczek-Geiringer, 1929].

**Power method over lower bounding Hessian of PNs**

We can employ IBP (Section 3.1) in order to obtain an expression of the lower bounding Hessian ($\boldsymbol{L_H}$) and evaluate Eq. (17) as:

$$\boldsymbol{L_H}\boldsymbol{v} = \frac{\mathcal{U}(\boldsymbol{H}_g(\boldsymbol{z}))\boldsymbol{v} + \mathcal{L}(\boldsymbol{H}_g(\boldsymbol{z}))\boldsymbol{v}}{2} + \left(\frac{\mathcal{L}(\boldsymbol{H}_g(\boldsymbol{z}))\mathbf{1} - \mathcal{U}(\boldsymbol{H}_g(\boldsymbol{z}))\mathbf{1}}{2}\right) * \boldsymbol{v}\,. \tag{18}$$

Applying IBP on Eq. (12) we obtain:

$$\begin{aligned}
\mathcal{L}(\boldsymbol{H}_g(\boldsymbol{z}))\boldsymbol{v} &= \sum_{i=1}^{k}(c_{ti} - c_{\gamma i})^+ \mathcal{L}(\nabla^2_{\boldsymbol{zz}}x_i^{(N)})\boldsymbol{v} + \sum_{i=1}^{k}(c_{ti} - c_{\gamma i})^- \mathcal{U}(\nabla^2_{\boldsymbol{zz}}x_i^{(N)})\boldsymbol{v} \\
\mathcal{U}(\boldsymbol{H}_g(\boldsymbol{z}))\boldsymbol{v} &= \sum_{i=1}^{k}(c_{ti} - c_{\gamma i})^- \mathcal{L}(\nabla^2_{\boldsymbol{zz}}x_i^{(N)})\boldsymbol{v} + \sum_{i=1}^{k}(c_{ti} - c_{\gamma i})^+ \mathcal{U}(\nabla^2_{\boldsymbol{zz}}x_i^{(N)})\boldsymbol{v}\,.
\end{aligned} \tag{19}$$

We can recursively evaluate $\mathcal{L}(\nabla^2_{\boldsymbol{zz}}x_i^{(n)})\boldsymbol{v}$ and $\mathcal{U}(\nabla^2_{\boldsymbol{zz}}x_i^{(n)})\boldsymbol{v}$ efficiently as these matrices can be expressed as a sum of rank-1 matrices as below.

**Proposition 1.** *Let* $\delta \in [\mathcal{L}(\delta), \mathcal{U}(\delta)]$ *be a real-valued weight, the matrix-vector products* $\mathcal{L}(\delta \cdot \nabla^2_{\boldsymbol{zz}}x_i^{(n)})\boldsymbol{v}$ *and* $\mathcal{U}(\delta \cdot \nabla^2_{\boldsymbol{zz}}x_i^{(n)})\boldsymbol{v}$ *can be evaluated as:*

$$\begin{aligned}
\mathcal{L}(\delta \cdot \nabla^2_{\boldsymbol{zz}}x_i^{(n)})\boldsymbol{v} &= \mathcal{L}(\delta \cdot \nabla_{\boldsymbol{z}}x_i^{(n-1)})\boldsymbol{w}_{[n]:i}^{+\top}\boldsymbol{v} + \mathcal{U}(\delta \cdot \nabla_{\boldsymbol{z}}x_i^{(n-1)})\boldsymbol{w}_{[n]:i}^{-\top}\boldsymbol{v} \\
&\quad + \boldsymbol{w}_{[n]:i}^{+}\mathcal{L}(\delta \cdot \nabla_{\boldsymbol{z}}x_i^{(n-1)\top})\boldsymbol{v} + \boldsymbol{w}_{[n]:i}^{-}\mathcal{U}(\delta \cdot \nabla_{\boldsymbol{z}}x_i^{(n-1)\top})\boldsymbol{v} \\
&\quad + \mathcal{L}(\delta'\nabla^2_{\boldsymbol{zz}}x_i^{(n-1)})\boldsymbol{v}\,,
\end{aligned} \tag{20}$$

$$\begin{aligned}
\mathcal{U}(\delta \cdot \nabla^2_{\boldsymbol{zz}}x_i^{(n)})\boldsymbol{v} &= \mathcal{L}(\delta \cdot \nabla_{\boldsymbol{z}}x_i^{(n-1)})\boldsymbol{w}_{[n]:i}^{-\top}\boldsymbol{v} + \mathcal{U}(\delta \cdot \nabla_{\boldsymbol{z}}x_i^{(n-1)})\boldsymbol{w}_{[n]:i}^{+\top}\boldsymbol{v} \\
&\quad + \boldsymbol{w}_{[n]:i}^{-}\mathcal{L}(\delta \cdot \nabla_{\boldsymbol{z}}x_i^{(n-1)\top})\boldsymbol{v} + \boldsymbol{w}_{[n]:i}^{+}\mathcal{U}(\delta \cdot \nabla_{\boldsymbol{z}}x_i^{(n-1)\top})\boldsymbol{v} \\
&\quad + \mathcal{U}(\delta'\nabla^2_{\boldsymbol{zz}}x_i^{(n-1)})\boldsymbol{v}\,,
\end{aligned} \tag{21}$$

*where* $\delta' \in [\mathcal{L}(\delta), \mathcal{U}(\delta)] \cdot [\mathcal{L}(\boldsymbol{w}_{[n]:i}^{\top}\boldsymbol{z} + 1), \mathcal{U}(\boldsymbol{w}_{[n]:i}^{\top}\boldsymbol{z} + 1)]$ *and vectors* $\mathcal{L}(\delta \cdot \nabla_{\boldsymbol{z}}x_i^{(n-1)})$ *and* $\mathcal{U}(\delta \cdot \nabla_{\boldsymbol{z}}x_i^{(n-1)})$ *can be obtained through IBP on Eq.* (13).

Lastly, by applying recursively Proposition 1 from $n = N$ to $n = 1$, starting with $\delta = 1$, we can substitute the results on Eq. (19) and then on Eq. (18) to efficiently evaluate a step of the power method (Eq. (17)) without needing to store the lower bounding Hessian matrix or needing to perform expensive matrix-vector products.

Overall, our lower bounding method consists in computing a valid value of $\alpha$ that satisfies that the $\alpha$-convexified objective $g_\alpha$ is convex, following Eqs. (4) and (5). In particular, we use $\alpha = \frac{\rho(\boldsymbol{L_H})}{2}$. $\rho(\boldsymbol{L_H})$ is computed via a power method, where the main operation $\boldsymbol{L_H}\boldsymbol{v}$ is evaluated without the need to compute or store the $\boldsymbol{L_H}$ matrix. Provided this valid $\alpha$, we perform PGD over $g_\alpha$ and this provides a lower bound of the global minima of Eq. (4).

### 3.4 Non-uniform diagonal shift

In order to obtain an estimate of the $\boldsymbol{\alpha}$ parameter as defined by Adjiman et al. [1998] in Eq. (7), we make use of the rank-1 matrices IBP rules defined in Appendix F.1. We also define the operator $\mathcal{M}(\cdot) = \max\{|\mathcal{L}(.)|, |\mathcal{U}(.)|\}$ and certain useful properties about it in Appendix F.2. Thanks to Lemmas 4 to 6, we can obtain an expression for $\mathcal{M}(\boldsymbol{H_z}(g))$. This will be used to compute the vector $\boldsymbol{\alpha}$ in the Non-uniform diagonal shift scenario for $\alpha$-convexification.

**Theorem 1.** *Let $f$ be a $N$-degree CCP PN defined as in Eq. (1). Let $g(\boldsymbol{z}) = f(\boldsymbol{z})_t - f(\boldsymbol{z})_\gamma$ for any $t \neq \gamma, t \in [o], \gamma \in [o]$. Let $\boldsymbol{H_g}(\boldsymbol{z})$ be the Hessian matrix of $g$, the operation $\mathcal{M}(\boldsymbol{H_g}(\boldsymbol{z}))$ results in:*

$$\mathcal{M}(\boldsymbol{H_g}(\boldsymbol{z})) \leq \sum_{i=1}^{k} |c_{ti} - c_{\gamma i}| \, \mathcal{M}\left(\nabla^2_{\boldsymbol{zz}} x_i^{(N)}\right) , \tag{22}$$

*where for $n = 2, ..., N$, we can express:*

$$\mathcal{M}\left(\nabla^2_{\boldsymbol{zz}} x_i^{(n)}\right) \leq \mathcal{M}\left(\nabla_{\boldsymbol{z}} x_i^{(n-1)}\right) |\boldsymbol{w}_{[n]i:}|^\top + |\boldsymbol{w}_{[n]i:}| \, \mathcal{M}\left(\nabla_{\boldsymbol{z}} x_i^{(n-1)}\right)^\top$$
$$+ \mathcal{M}\left(\boldsymbol{w}_{[n]i:}^\top \boldsymbol{z} + 1\right) \mathcal{M}\left(\nabla^2_{\boldsymbol{zz}} x_i^{(n-1)}\right) . \tag{23}$$

*Lastly, for $n = 1$, $\mathcal{M}\left(\nabla^2_{\boldsymbol{zz}} x_i^{(1)}\right) = \boldsymbol{0}_{d \times d}$ is a $d \times d$ matrix full of zeros.*

As one can observe, in Theorem 1, the matrix $\mathcal{M}(\boldsymbol{H_g}(\boldsymbol{z}))$ is expressed as a sum of rank-1 matrices. This allows to efficiently compute $\sum_{j \neq i} \mathcal{M}(h_g(\boldsymbol{z})_{ij}) \frac{d_i}{d_j}$, which is necessary for the right term in Eq. (7). For the left term in Eq. (7), we can efficiently compute the lower bound of the diagonal of the Hessian matrix by using the rules present in Section 3.1 and Appendix F.1.

## 4 Experiments

In this Section we show the efficiency of our method by comparing against a simple Black-box solver. Tightness of bounds is also analyzed in comparison with IBP and DeepT-Fast [Bonaert et al., 2021], a zonotope based verification method able to handle multiplications tighter than IBP. Finally, a study of the performance of our method in different scenarios is performed. Unless otherwise specified, every network is trained for 100 epochs with Stochastic Gradiend Descent (SGD), with a learning rate of $0.001$, which is divided by 10 at epochs $[40, 60, 80]$, momentum 0.9, weight decay $5 \cdot 10^{-5}$ and batch size 128. We thoroughly evaluate our method over the popular image classification datasets MNIST [LeCun et al., 1998], CIFAR10 [Krizhevsky et al., 2014] and STL10 [Coates et al., 2011]. Every experiment is done over the first 1000 images of the test dataset, this is a common practice in verification [Singh et al., 2019]. For images that are correctly classified by the network, we sequentially verify robustness against the remaining classes in decreasing order of network output. Each verification problem is given a maximum execution time of 60 seconds, we include experiments with different time limits in Appendix E. Note that the execution time can be longer as execution is cut in an asynchronous way, i.e., after we finish the iteration of the BaB algorithm where the time limit is reached. All of our experiments were conducted on a single GPU node equipped with a 32 GB NVIDIA V100 PCIe.

### 4.1 Comparison with a Black-box solver

In this experiment, we compare the performance of our BaB verification algorithm with the Black-box solver Gurobi [Gurobi Optimization, LLC, 2022]. Gurobi can globally solve Quadratically Constrained Quadratic Programs whether they are convex or not. As this solver cannot extend to higher degree polynomial functions, we train $2^{\text{nd}}$ degree PNs with hidden size $k = 16$ to compare the verification time of our method with Gurobi. In order to do so, we express the verification objective as a quadratic form $g(\boldsymbol{z}) = f(\boldsymbol{z})_t - f(\boldsymbol{z})_a = \boldsymbol{z}^\top \boldsymbol{Q} \boldsymbol{z} + \boldsymbol{q}^\top \boldsymbol{z} + c$ this together with the input constraints $\boldsymbol{z} \in [\boldsymbol{l}, \boldsymbol{u}]$ is fed to Gurobi and optimized until convergence.

The black-box solver approach neither scales to higher-dimensional inputs nor to higher polynomial degrees. With this approach we need $\mathcal{O}(d^2)$ memory to store the quadratic form, which makes it unfeasible for datasets with higher resolution images than CIFAR10. On the contrary, as seen in Table 1, our approach does not need so much memory and can scale to datasets with larger input sizes like STL10.

Table 1: Verification results for $2^{nd}$ degree PNs. Columns #F, #T and #t.o. refer to the number of images where robustness is falsified, verified and timed-out respectively. When comparing with a black-box solver, our method is much faster and can scale to higher dimensional inputs. This is due to our efficient exploitation of the low-rank factorization of PNs.

| Dataset | Model | Correct | $\epsilon$ | VPN (Our method) | | | | Gurobi | | | |
|---|---|---|---|---|---|---|---|---|---|---|---|
| | | | | time | F | T | t.o. | time | F | T | t.o. |
| MNIST | $2 \times 16$ | 961 | 0.00725 | **1.76** | 37 | 924 | 0 | 16.6 | 37 | 924 | 0 |
| $(1 \times 28 \times 28)$ | | | 0.013 | **1.78** | 71 | 890 | 0 | 15.13 | 71 | 890 | 0 |
| | | | 0.05 | **1.43** | 682 | 267 | 12 | 6.25 | 691 | 270 | 0 |
| | | | 0.06 | **1.5** | 790 | 155 | 16 | 4.47 | 799 | 162 | 0 |
| CIFAR10 | $2 \times 16$ | 460 | 1/610 | **1.03** | 90 | 370 | 0 | 328.0 | 90 | 370 | 0 |
| $(3 \times 32 \times 32)$ | | | 1/255 | **1.0** | 183 | 277 | 0 | 250.07 | 183 | 277 | 0 |
| | | | 4/255 | **0.92** | 427 | 28 | 5 | 87.93 | 429 | 31 | 0 |
| STL10 | $2 \times 16$ | 362 | 1/610 | **5.06** | 142 | 220 | 0 | | | | |
| $(3 \times 96 \times 96)$ | | | 1/255 | **3.61** | 246 | 113 | 3 | | out of memory | | |
| | | | 4/255 | **1.39** | 360 | 1 | 1 | | | | |

## 4.2 Comparison with IBP and DeepT-Fast

In this experiment we compare the tightness of the lower bounds provided by IBP, DeepT-Fast and $\alpha$-convexification and their effectiveness when employed for verification. This is done by executing one upper bounding step with PGD and one lower bounding step for each lower bounding method over the initial feasible set provided by $\epsilon$ (see Eq. (3)). We compare the average of the distance from each lower bound to the PGD upper bound over the first 1000 images of the MNIST dataset for PNs with hidden size $k = 25$ and degrees ranging from 2 to 7. We also evaluate verified accuracy of $2^{nd}$ (PN_Conv2) and $4^{th}$ (PN_Conv4) PNs with IBP, DeepT-Fast and $\alpha$-convexification in the Uniform diagonal shift setup. We employ a maximum time of 120 seconds. For details on the architecture of these networks, we refer to Appendix E.

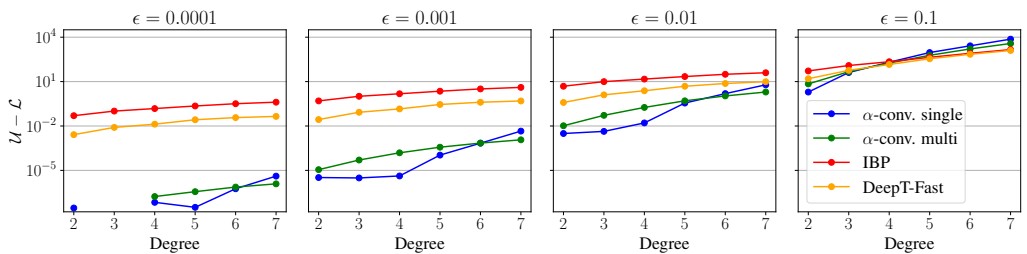

Figure 3: Average difference in log-scale between PGD upper bound ($\mathcal{U}$) and lower bound ($\mathcal{L}$) provided by BP (red), DeepT-Fast [Bonaert et al., 2021] (orange), $\alpha$-convexification with Uniform diagonal shift (blue) and $\alpha$-convexification with Non-uniform diagonal shift (green) of the first 1000 images of the MNIST dataset. $\alpha$-convexification bounds are significantly tighter than IBP and DeepT-Fast for small $\epsilon$ values and all PN degrees from 2 to 7.

When using IBP, we get a much looser lower bound than with $\alpha$-convexification, see Fig. 3. Only for high-degree, high-$\epsilon$ combinations IBP lower bounds are closer to the PGD upper bound. In practice, this is not a problem for verification, as for epsilons in the order of $0.1$, it is really easy to find adversarial examples with PGD and there will be no accuracy left to verify. DeepT-Fast significantly outperforms IBP bounds across all degrees and $\epsilon$ values. But, as observed in Fig. 3, except for big $\epsilon$ values, its performance is still far from the one provided by both $\alpha$-convexification methods. When comparing both $\alpha$-convexification methods (blue and green lines in Fig. 3), we observe that for small degree PNs ($N < 5$), in the Uniform diagonal shift case we are able to obtain tighter bounds.

The looseness of the IBP lower bound is confirmed when comparing the verified accuracy with IBP and the rest of lower bounding methods, see Table 2. With $\alpha$-convexification, we are able to verify the accuracy of $2^{nd}$ and $4^{th}$ order PNs almost exactly (almost no gap between the verified accuracy and its upper bound) in every studied dataset, while with IBP, we are not able to verify robustness for a single image in any network-$\epsilon$ pair, confirming the fact that IBP cannot be used

Table 2: Verification results with our method employing IBP, DeepT-Fast and $\alpha$-convexification for lower bounding the objective. `Acc.%` is the clean accuracy of the network, `Ver.%` is the verified accuracy and `U.B.` its upper bound. When using $\alpha$-convexification bounds we get verified accuracies really close to the upper bound, while when using IBP verified accuracy is $0$ for every network-$\epsilon$ pair, which makes it unsuitable for PN verification.

| Dataset | Model | Acc.% | $\epsilon$ | IBP | | DeepT-Fast [Bonaert et al., 2021] | | VPN (ours) ($\alpha$-convexification) | | U.B. |
|---|---|---|---|---|---|---|---|---|---|---|
| | | | | Time(s) | Ver.% | Time(s) | Ver.% | Time(s) | Ver.% | |
| MNIST | PN_Conv4 | 98.6 | 0.015 | 0.3 | 0.0 | 2.3 | 91.3 | 50 | **96.3** | 96.4 |
| | | | 0.026 | 0.4 | 0.0 | 3.7 | 59.7 | 69 | **92.9** | 94.8 |
| | | | 0.3 | 0.6 | 0.0 | 0.7 | 0.0 | 13.8 | 0.0 | 0.0 |
| CIFAR10 | PN_Conv2 | 63.5 | 1/255 | 0.3 | 0.0 | 2.0 | 23.3 | 136.2 | **44.4** | 44.6 |
| | | | 2/255 | 0.5 | 0.0 | 0.6 | 1.4 | 89.2 | **25.4** | 27.5 |
| | PN_Conv4 | 62.6 | 1/255 | 0.4 | 0.0 | 2.2 | 19.5 | 274.6 | **45.5** | 46.7 |
| | | | 2/255 | 0.5 | 0.0 | 0.5 | 0.5 | 224.1 | **16.5** | 30.5 |
| STL10* | PN_Conv4 | 38.1 | 1/255 | 3.4 | 0.0 | 26.0 | 14.7 | 2481.0 | **21.7** | 21.9 |

* Results obtained in the first 360 images of the dataset due to the longer running times because of the larger input size of STL10.

for PN verification. The improvements in the bounds when utilizing DeepT-Fast instead of IBP is clearly seen in verification results in Table 2. With DeepT-Fast, we are able to effectively verify PNs faster than with $\alpha$-convexification, but achieving a much lower verified accuracy (`Ver%`) than with $\alpha$-convexification. As a reference, for CIFAR10 PN_Conv2 at $\epsilon = 2/255$, with $\alpha$-convexification we obtain $25.4\%$ verified accuracy, while with DeepT-Fast, we can just obtain $1.4\%$ verified accuracy. It is worth highlighting that DeepT-Fast can also scale to verify networks trained on STL10.

## 5   Conclusion

We propose a novel $\alpha$-BaB global optimization algorithm to verify polynomial networks (PNs). We exhibit that our method outperforms existing methods, such as black-box solvers, IBP and DeepT-Fast [Bonaert et al., 2021]. Our method enables verification in datasets such as STL10, which includes RGB images of $96 \times 96$ resolution. This is larger than the images typically used in previous verification methods. Note that existing methods like IBP and DeepT-Fast are also able to scale to STL10 buth with a lower verified accuracy. Our method can further encourage the community to extend verification to a broader class of functions as well as conduct experiments in datasets of higher resolution. We believe our method can be extended to cover other twice-differentiable networks in the future.

### Limitations

As discussed in Appendix E.2, our verification method does not scale to high-degree PNs. Even though we can verify high-accuracy PNs (see Table 2), we are still far from verifying the top performing deep PNs studied in Chrysos et al. [2021]. Another problem that we share with ReLU NN verifiers is the scalability to networks with larger input size [Wang et al., 2021]. In this work we are able to verify networks trained in STL10 [Coates et al., 2011], but these networks are shallow, yet their verification still takes a long time, see Table 2.

## Acknowledgements

We are deeply thankful to the reviewers for providing constructive feedback. Research was sponsored by the Army Research Office and was accomplished under Grant Number W911NF-19-1-0404. This project has received funding from the European Research Council (ERC) under the European Union's Horizon 2020 research and innovation programme (grant agreement number 725594 - time-data). This work was supported by the Swiss National Science Foundation (SNSF) under grant number 200021_178865. This project has received funding from the European Research Council (ERC) under the European Union's Horizon 2020 research and innovation programme (grant agreement n° 725594 - time-data). This work was supported by Zeiss. This work was supported by SNF project – Deep Optimisation of the Swiss National Science Foundation (SNSF) under grant number 200021_205011.

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
