# OpenReview forum: "Sound and Complete Verification of Polynomial Networks"
_NeurIPS.cc/2022/Conference — NeurIPS 2022 Accept_

### Official Review · Reviewer_sZV9 · 2022-06-24

**Rating:** 7
**Confidence:** 4
**Soundness:** 3 good
**Presentation:** 3 good
**Contribution:** 3 good

**Summary:**

This work introduces a method to prove robustness of polynomial networks against $l^\infty$ perturbations. The method relies upon finding multivariate polynomials of degree 2 called $f_\alpha$ lower bounding the differences in the classification score $g(z) = f(z)_t - f(z)_a$. The lower binding function $f_\alpha$ relies on the parameter $\alpha$ which is estimated via a combination of branch and bound and interval arithmetic. Crucially, the approach relies on the twice differentiable structure of polynomial networks and the combination with branch and bound over the input make the method complete. The approach is evaluated on MNIST, CIFAR10 and STL10 and compared against standard of the shelf solvers (Gurobi) as well as interval arithmetic. The code was provided together with the submission.

**Questions:**

- Have the authors considered the possibility to split the sum in Eq. 5, thus maybe enabling different $\alpha$ to increase precision of the approach? Would this be possible? If yes, how would the runtime complexity change?
- In Eq. 13: how is A defined?
- In Eq. 15: usually for the power method, the sequence is defined by $v_k = \tfrac{M v_{k-1}}{\| M v_{k-1} \|}$. Is there a specific reason to have $M$ twice as in Eq. 15?
- The power method numerical, thus has an approximation error. Is this approximation error handled soundly for the later analysis? Similarly, in Line 127-129, for PGD also there is a numerical approximation issue.
- Do I assume correctly that the method is not floating point sound? (I think this is not central but I am curious anyhow)
- Can more complex networks used for CIFAR10 and STL10 that offer higher precision?
- Please speculate: Can and if how can you use your certification method to train the polynomial network?
- Could the input splitting technique be improved by i.e. weighting the interval distance in each dimension by the diagonal entry of the hessian? (i.e. weighting by the second derivatives of the dimension in question?)
- in Line 110: are the sets really disjoint? Wouldn’t the interval [1,3] be spliced in [1,2] and [2,3] having non-empty intersection?

**Limitations:**

The limitations are appropriately addressed.

**Strengths And Weaknesses:**

Verifying new architectures is a challenging, interesting and important field of research. Polynomial networks show strong performance along a range of tasks, justifying the authors interest. The paper is mostly well written and the notation is clearly explained. Further, the Limitations are stated clearly in the section for this purpose. The might also be applicable to ReLU networks, as the authors state. Input refinement based approach to make sound verification complete where also considered earlier (see https://arxiv.org/pdf/1804.10829.pdf, https://arxiv.org/abs/1809.08098), the authors may want to have a look.

The approach does not seem to scale to large models or challenging datasets, which is also clearly stated by the authors. However, it would have been interesting, so see some larger (more accurate) models if feasible for CIFAR10 and STL10 in order to increase the number of correct classified inputs. Further, the hardware (GPU) is not specified. A comparison to other complete & non complete methods (evaluated on other networks) would have given an indication how polynomial networks with this certification compares to say standard ReLU convolutional networks performing the same task. Some details of the approach remained unclear, an explicit example illustrating the method would have been helpful.

## After the rebuttal:
The authors clarified all my questions and provided in the general response an informative comparison, solidifying their evaluation. I think that this paper presents new and useful ideas for the field. I recommend acceptance.

---

> ### Author Response · Authors · 2022-08-01
> **Response to reviewer sZV9**
>
> We thank reviewer sZV9 for their review and meaningful suggestions. We answer the questions below:
>
> * **Q1**: Are more accurate/larger PNs possible to verify?
>
> More accurate PNs do exist (Chrysos et al., 2021), but these networks are built by stacking blocks of PNs, which is known as a product of polynomials. In this work, we provide theoretical results for estimating $\alpha$ in the product of polynomials case, see Proposition 2 in the supplementary. However, in our preliminary experiments we observed that this $\alpha$ estimate led to quite loose lower bounds, not allowing the verification of these architectures for now.
>
> ____
> * **Q2**: Lack of comparison with other methods.
>
> We added a comparison with DeepT-Fast (Bonaert et al., 2021) in our [general response](https://openreview.net/forum?id=gsdHDI-p6NI&noteId=HpWOZg6Px5M) and in the manuscript, our method proved to be the best in PN verification.
>
> ____
> * **Q3**: Difficulties to understand VPN.
>
> We understand the difficulties in understanding our method and thank reviewer sZV9 for highlighting it. A schematic of our method is available in Fig. 2 (https://imgur.com/a/Oxot4rK), note that $\alpha$ estimation is done outside of the Branch and Bound procedure.
>
> ____
> * **Q4**: Have the authors considered the possibility of splitting the sum in Eq. 5?
>
> Indeed, this can be done. We cover this answer in the [general response](https://openreview.net/forum?id=gsdHDI-p6NI&noteId=HpWOZg6Px5M).
>
> ____
> * **Q5**: In Eq. 13: how is A defined?
>
> Thanks for pointing out this typo, and it should be revised as $U(M)$, as shown in the manuscript.
>
> ____
> * **Q6**: Why do you perform two steps of the power method per iteration?
>
> This is done in order to deal with negative dominant eigenvalues. In the case at iteration $n$, the vector is already the dominant eigenvector with eigenvalue $\lambda < 0$, a regular power method step will lead to $\mathbf{v}\_{n+1} = \frac{\mathbf{M}\mathbf{v}\_{n}}{||\mathbf{M}\mathbf{v}\_{n}||\_{2}} = \frac{\lambda \mathbf{v}}{|\lambda|} = -\mathbf{v}\_{n}$. Check that the vector is flipped and the stopping criterion will be $||\mathbf{v}\_{n+1} - \mathbf{v}\_{n}||\_{2} = 2\lambda$ entering in an infinite loop. By computing an extra step of the power method, we flip the vector again and are able to finish the algorithm correctly. For more details on this method, we refer to Algorithm 2 in the supplementary.
>
> ____
> * **Q7**: How do you deal with approximation errors in numerical methods (power method and PGD)?
>
> We empirically observed that our stopping conditions are sufficient and only a negligible error of $10^{-5}$ remains. For our power method, the stopping condition is that the $\ell_2$ norm of the difference between eigenvector estimations between steps must be smaller than 10^{-5} ($||\mathbf{v}\_{n+1} - \mathbf{v}\_{n}||\_{2} < 10^{-5}$). Such a small error margin in the eigenvector estimation forces the power method to perform many steps, but leads to a really accurate estimation of the spectral norm. Similarly, for PGD we take an error margin of $10^{-5}$, ensuring a really accurate estimation of the lower bound.
> ____
> * **Q8**: Is VPN floating point sound?
>
> No, it is not.
> ____
> * **Q9**: Speculate on how to integrate your method in the PN training.
>
> This is an interesting question. We contemplate two possible approaches:
> 1) Use our method to find harder adversarial examples than the ones found with PGD. These adversarial examples could then be used to augment the training data and **improve robustness**.
> 2) Employ our $\alpha_i$ estimation method (see [general response](https://openreview.net/forum?id=gsdHDI-p6NI&noteId=HpWOZg6Px5M)) to every certain number of epochs/minibatches introduce a penalization term in the loss function involving the norm of the $\mathbf{\alpha}$ vector. Penalizing the norm of this vector will most likely **ease the verification** of PNs, as the lower the norm of $\mathbf{\alpha}$, the tighter the bounds provided by $\alpha$-convexification.
> However, we presume this will be very expensive to do.
> ____
> * **Q10**: New input set splitting strategy suggestion.
>
> Thanks for this interesting suggestion, we elaborate on this topic in our [general response](https://openreview.net/forum?id=gsdHDI-p6NI&noteId=lpfY4tMV1t3x).
> ____
> * **Q11**: In Line 110: are the sets really disjoint?
>
> They are not. Thanks for mentioning this fact. Theoretically, for the BaB procedure to converge, we don’t need both sets to be disjoint, e.g., [1,3] could be split into [1,2.5] and [1.5,3]. What we do need is the intervals to be closed for performing PGD. We have amended this in line 110.
> ____
> We hope that our answers have addressed the questions of the reviewer; we would be happy to elaborate further if something remains unclear. Once again, we appreciate the thoughtful feedback of the reviewer.
>
> ### References
> Chrysos et al., Deep Polynomial Neural Networks, TPAMI 2021.
>
> Bonaert et al., Fast and precise certification of transformers, PLDI 2021.

---

> > ### Author Response · Authors · 2022-08-01
> > **Response on GPU hardware used**
> >
> > * **Q12**: Hardware (GPU) is not specified.
> >
> > All of our experiments were performed in a **single** node with a 32 GB NVIDIA V100 PCIe GPU. We specified the brand and model of the GPU in the new version in lines 241-242.

---

> > > ### Author Response · Authors · 2022-08-06
> > > **Have the concerns of the reviewer been addressed?**
> > >
> > > Dear reviewer sZV9,
> > >
> > > We are thankful for your considerate questions. We agree with the reviewer that scalability to larger models might be an issue. We believe this is a common concern with neural network verification. Our method is the first one to make a step towards verification  of Polynomial Networks and as such, we expect that future work will build on top of our work and enable such non-trivial extensions to more complex models. To achieve that, we have made several new observations on verification of this class of functions.
> > >
> > > We would like to check if the reviewer has been covered by our explanation, and the improvements made to the manuscript based on your suggestions as well as the [new empirical evidence](https://openreview.net/forum?id=gsdHDI-p6NI&noteId=HpWOZg6Px5M).

---

> > > ### Comment · Reviewer_sZV9 · 2022-08-09
> > > **Comments**
> > >
> > > Dear Authors
> > >
> > > thank you very much for your very detailed and informative response, particularly the comparison against DeepT-Fast, nicely indicating some of the tradeoffs the two methods have. All of my concerns have been sufficiently addressed by the authors. I will update my score accordingly.

---

> > > > ### Author Response · Authors · 2022-08-09
> > > > **Thank you for your response**
> > > >
> > > > Dear reviewer sZV9,
> > > >
> > > > We are grateful for your constructive feedback and your acknowledgement of our updates.

---

### Official Review · Reviewer_bVTt · 2022-07-10

**Rating:** 5
**Confidence:** 4
**Soundness:** 2 fair
**Presentation:** 2 fair
**Contribution:** 2 fair

**Summary:**

This paper is a direct application of complete formal verification  against robustness applied to polynomial networks. They propose a new efficient computation of the Hessian bound in order to guarantees property wit the BaB method (a recent MILP verification method). They applied their method to 3 datasets, MNIST, CIFAR10 and STL10 for 1 model and different noise level +  comparison with IBP for multiple PN models on MNIST.

**Questions:**

Explicit the last point of weakness (Poor explications of why PN is an interesting family to explore (compared to BNN for example) in term of theory or application (NLP - graphNN - image generation...)) can improve my opinion on the paper.
But I think the verifiable accuracy is too low compared to state of the art.
However, if there is a clear proof of PN model application for industry then

**Limitations:**

cf. last point on question.

**Strengths And Weaknesses:**

Strengths:
Novelty of the approach: complete verification of DNN specific family
Important topic
Sound proof of the Hessian bounds
Clarity of the paper
Very fast verification

Weaknesses
Poor experiments comparison: Not comparison with SAT based method, no comparison with other  BaB methods. I understand it is not PN based verification but still important (https://arxiv.org/abs/2005.03597 and https://arxiv.org/abs/2103.06624). For example evry verification paper gives the resultas for 0.3 MNST and 2/255 for cfar10
Poor verifiable accuracy results compared to results above.
Poor explications of why PN is an interesting family to explore (compared to BNN for example) in term of theory or application (NLP - graphNN - image generation...)

---

> ### Author Response · Authors · 2022-08-01
> **Response to reviewer bVTt**
>
> We thank reviewer bVTt for their review. We answer the questions below:
>
> - **Q1**: Lack of comparison with SAT solvers and BaB MILP solvers.
>
> Let us firstly clarify why our optimization problem cannot be expressed as a MILP. **Our problem, present in Eq. 4, is a non-linear program with linear constraints**. In ReLU NN verification, ReLU layers can be encoded with the help of binary variables and pre-activation bounds, reducing the objective to linear and overall to a MILP problem, see (Tjeng et al., 2019). This is not the case of PNs, which achieve non-linearity with Hadamard products and not piecewise linear activation functions. Therefore, BaB MILP solvers as $\beta$-CROWN cannot be used for PN verification.
>
> Then regarding SAT solvers for PN verification, in (Jia and Rinard, 2020), they are able to use these solvers due to the binary nature of BNN. SMT solvers, where linear inequalities are also allowed as clauses, were initially used for ReLU NN verification, see Reluplex (Katz et al., 2017). In the case of PNs, similarly to how expression as a MILP is not plausible, we cannot encode the behavior of the network with linear inequalities (SMT problem) and clearly not binary variables (SAT problem), this is due to Hadamard products present in PNs.
>
> However, during this rebuttal, we conducted experiments with the zonotope based verification method of DeepT-Fast (Bonaert et al., 2021). See [general response](https://openreview.net/forum?id=gsdHDI-p6NI&noteId=HpWOZg6Px5M) for more details.
>
> ____
> - **Q2**: No results for MNIST networks at $\epsilon = 0.3$
>
> According to your suggestions, we have already conducted experimental results on $\epsilon = 0.3$ for MNIST. We already had results for CIFAR10 at $\epsilon = 2/255$. Everything is reported by Tab. 2.
> In the future, we will consider using defense mechanisms with such large $\epsilon$ values, similarly to the literature (Singh et al., 2019). We thank the reviewer for pointing this out.
>
> ____
> - **Q3**: Why are PNs an interesting family to explore?
>
> There is ample work on Polynomial Networks, for instance (Rendle, 2010; Jayakumar et al., 2019; Chrysos et al., 2021b; Choraria et al., 2022). Benefits from using this class of functions have been demonstrated in both theory and empirical results:
>
> * Theoretical results: more expressive (Rendle, 2010) with different spectral bias that results in learning higher frequency functions firstly (Choraria et al., 2022). Also, in scenarios where only additions and multiplications can be used, like (Brakerski et al., 2014), PNs are a natural fit.
>
> * Empirical results: State-of-the-art in the challenging task of face recognition (https://paperswithcode.com/sota/face-verification-on-megaface), or in non-euclidean representation learning (Chrysos et al., 2021a).
>
> We emphasize that these are not results in our work, but they are motivating points for studying polynomials. Our goal here is to provide a verification algorithm that can provide certificates for this class of functions, since none existed previously.
>
> ____
> - **Q4**: Comparison with neural networks:
>
> Regarding the low verified accuracy, we remark that it is not the objective of this paper to analyze the robustness of PNs in comparison with other architectures, since it is hard to compare them. As we explain above, the class of polynomial nets has already demonstrated success. In addition, it does not have many common elements with NN verifiers, therefore it is hard to use the same verifiers used in the literature, e.g. beta-crown etc, as suggested by the reviewer. *If the reviewer has any concrete insights on how this is possible, we will be happily including a related comparison in Table 2*.
>
> ____
> We hope that our answers have addressed the questions of the reviewer both in terms of our paper and PNs; we would be happy to elaborate further if something remains unclear.
>
> ### References
> Chrysos et al., Deep Polynomial Neural Networks, TPAMI 2021.
>
> Rendle, Factorization Machines, ICDM 2010.
>
> Jayakumar et al., Multiplicative Interactions and Where to Find Them, ICLR 2020.
>
> Chrysos et al., Polynomial Networks in Deep Classifiers, arXiv 2021
>
> Brakerski et al., (Leveled) Fully Homomorphic Encryption without Bootstrapping, ACM Transactions on Computation Theory 2014.
>
> Choraria et al., The Spectral Bias of Polynomial Neural Networks, ICLR 2022.
>
> Tjeng et al., Evaluating Robustness of Neural Networks with Mixed Integer Programming, ICLR 2019.
>
> Jia and Rinard, Evaluating Robustness of Neural Networks with Mixed Integer Programming, NIPS 2020.
>
> Katz et al., Reluplex: An Efficient SMT Solver for Verifying Deep Neural Networks, CAV 2017.
>
> Bonaert et al., Fast and precise certification of transformers, PLDI 2021.
>
> Sing et al., Beyond the Single Neuron Convex Barrier for Neural Network Certification, NIPS 2019.

---

> > ### Comment · Reviewer_bVTt · 2022-08-08
> > **Noted**
> >
> > Noted, thanks, I upgrade the ark accordingly. But I still think that you are too far away from SoA. You should maybe show taht you can verify others properties.

---

> > > ### Author Response · Authors · 2022-08-09
> > > **Response to the state-of-the-art claims**
> > >
> > > We thank reviewer bVTt for their time in reviewing our work and acknowledging our effort to address the concerns. We answer your remaining concerns bellow:
> > >
> > > > Still far away from the State of the Art (SotA).
> > >
> > > We agree with the reviewer that the verified accuracies do not match the ones for top performing Neural Networks. Nevertheless, improving robustness of PNs is beyond the scope of this paper. Alternatively, we compare our method with other promising verification methods in the PN verification task.
> > >
> > > As we indicate in our [previous response](https://openreview.net/forum?id=gsdHDI-p6NI&noteId=Gu0gZueeRZL) applying the NN verification methods for verification of Polynomial Nets (PNs) and other architectures is *non-trivial*. For instance,  (Narodytska et al., 2018; Shi et al., 2020) do not include comparisons with NN verifiers, since the class they are focusing on differs from the standard NNs used in NN verification. Their comparison is limited to simple baselines like IBP (Shi et al., 2020) or SAT/SMT/MILP solvers (Narodytska et al., 2018).
> > >
> > > We agree with the reviewer that providing strong baselines is an important aspect. To that end, we had originally conducted experiments with IBP *and* with SotA QCQP solver (Gurobi). Upon the suggestion of the [reviewer rPvn](https://openreview.net/forum?id=gsdHDI-p6NI&noteId=IkPaYACYA1I), we have also included DeepT-Fast (Bonaert et al., 2021) as a baseline. When comparing our method against DeepT-Fast we find that **we can obtain tighter lower bounds and higher verified accuracies**. See Tab. 2 in the manuscript and [general response](https://openreview.net/forum?id=gsdHDI-p6NI&noteId=HpWOZg6Px5M). Given those baselines in the context of PNs, we demonstrate how our method outperforms those baselines and obtains the **best verification results for PNs**.
> > >
> > > If the reviewer has any other methods that can be used in the context of PNs, we will analyze those and include a discussion in the final version of our work.
> > >
> > > ### References
> > >
> > > Narodytska et al., Verifying Properties of Binarized Deep Neural Networks, AAAI 2018
> > >
> > > Shi et al., Robustness Verification for Transformers, ICLR 2020
> > >
> > > Bonaert et al., Fast and precise certification of transformers, PLDI 2021.

---

### Official Review · Reviewer_rPvn · 2022-07-11

**Rating:** 7
**Confidence:** 5
**Soundness:** 4 excellent
**Presentation:** 3 good
**Contribution:** 4 excellent

**Summary:**

The paper considers developing a sound and complete verification method for robustness certification of polynomial networks (PNs) against norm-based perturbations. The verifier developed here combines input splitting with $\alpha$-convexification. The main algorithmic contribution is in computing this convexification which involves interval bound propagation (IBP) to bound the lower bounding hessian $L_H$ of the PNs and employing a standard power method to compute the $\alpha$ on the $L_H$. Comparisons against Gurobi and IBP show that the proposed method can enable more precise and scalable verification.

**Questions:**

* On line 119, $\lambda_{min}$ is undefined (I assume its minimum eigenvalue). In eq. (5) how is $z_i$ related to $z$ (I assume these are components of z)? What is $\mathcal{U}(A)$ in (13)?

* In Section 3.3, how is the initial vector $v_0$ chosen? Does the initialization affect the speed of the method? Do you always converge or using a value before convergence can still lead to sound verification?

* The input splitting strategy used by your algorithm appears suboptimal. There are better strategies available in the literature that can improve the results of your method (e.g., Efficient Verification of ReLU-Based Neural Networks via Dependency Analysis, AAAI 2020). Any particular reason why you chose the suboptimal strategy?

* How difficult it is to compute the lower bound of the eigenvalue of the Hessian with relational analysis such as DeepZ/Fast-Lin or DeepPoly/CROWN?

* For ReLU-based networks, do you need to use Clark jacobians for your method to work as done in https://dl.acm.org/doi/abs/10.1145/3498718, or non-differentiability at x=0 does not affect your method? If it does not, then why?

* "We take advantage of the low-rank decomposition characterizing PNs to efficiently perform a power method over the lower bounding Hessian." so your method works for only PNs with this property? If yes, then an extension to regular NN with Sigmoid/Tanh activation may not be possible?

**Limitations:**

The authors adequately addressed the limitations of their method in Section 6. The proposed work has no potential negative societal impact.


**Strengths And Weaknesses:**

Strengths:

+ The problem considered here is novel and well-motivated. There is not much work on the certification of polynomial networks which have also been shown to be vulnerable to adversarial attacks.

+ While $\alpha$-convexification and IBP are known techniques in the literature, their combination has not been applied for NN verification before to the best of my knowledge. Further, the proposed framework is general and can be extended to other neural network architectures with twice differentiable activation functions.

+ While there are some issues with the writing that I point out later, I enjoyed reading the paper overall. It is well-organized and relatively easy to follow.


Weakness:

- Since the primary operation in PNs is multiplication, I am not sure why the authors cannot compare with other methods in the literature (e.g., https://dl.acm.org/doi/abs/10.1145/3453483.3454056, https://openreview.net/forum?id=BJxwPJHFwS) that can handle multiplication more precisely than IBP. In my opinion, IBP is a very weak verifier for networks that are not trained to be easy to verify. Adding comparisons against these baselines will make the evaluation more compelling. Note I am not so concerned about the size of the networks/datasets. In many real-world applications, the small NNs are trained on datasets with much smaller dimensionality than STL10 (e.g., see security classifiers here: https://arxiv.org/abs/2105.11363, or the ones in Bio https://ieeexplore.ieee.org/abstract/document/9680252), therefore even though your verifier does not scale to large Vision/NLP datasets yet, there is still some practical value to your work. In fact, I also encourage the authors to consider these domains for their method in the future to enable more real-world impact of their work instead of focusing only on vision/NLP models which are often limited to academic/industrial competition and still not widely employed in the safety-critical settings due to their brittleness.

---

> ### Author Response · Authors · 2022-08-01
> **Response to reviewer rPvn**
>
> We thank the reviewer rPvn for such a thorough review. We answer the questions below:
>
> - **Q1**: Comparison with DeepT-Fast.
>
> We include your suggestion for comparison with DeepT-Fast more information is provided in the [general response](https://openreview.net/forum?id=gsdHDI-p6NI&noteId=HpWOZg6Px5M).
>
> ____
> - **Q2**: Undefined symbols and small typos.
>
> We thank the reviewer for pointing out the typos, see (see lines 120 and 122 and Eq. 15). Concretely:
>
> The notation $\lambda_{\text{min}}$ is the minimum eigenvalue.
> The notation $z_i$ is the $i$-th component of $\mathbf{z}$, as specified in lines 50-56.
> $\mathcal{U}(\mathbf{A})$ is a typo, which should be revised as $\mathcal{U}(\mathbf{M})$.
>
> ____
> - **Q3**: Initialization of $\mathbf{v}_{0}$ and convergence of power method.
>
> Each element in the initial vector $\mathbf{v}\_{0}$ is uniformly sampled from [0,1], and then $\ell_2$ normalized. Different initializations do not affect the final solution but have different time costs for convergence.  We have clarified this in lines 654-656. Our power method needs to be run until convergence to ensure the resulting $\alpha$ guarantees convexity of $g_{\alpha}$.
>
> ____
> - **Q4**: Suboptimal splitting strategy.
>
> Thanks for highlighting this aspect, we elaborate on this topic in our [general response](https://openreview.net/forum?id=gsdHDI-p6NI&noteId=lpfY4tMV1t3x).
>
> ____
> - **Q5**: Is it easy to compute $\alpha$ with DeepZ/Fast-Lin or DeepPoly/CROWN?
>
> We consider that it is non-trivial to do. In our method, we use IBP for computing the bounds of the Hessian matrix and perform a power method over the lower bounding Hessian. In our case, the IBP bounds allow expressing the lower bounding Hessian as a sum of rank-1 matrices, which allows the implementation of an efficient power method.
>
> Any method for computing sound bounds of the Hessian matrix could be used. For example, one could theoretically use DeepZ to build a zonotope abstraction of the elements in the Hessian matrix, and then use this for computing the lower bounding Hessian and its spectral radius. Our intuition is this will not lead to such an efficient implementation of the power method. But, similarly to how going from using IBP to DeepT-Fast, greatly improves the bounds and allows PN verification (see general response), going from IBP to a zonotope based method for bounding the Hessian and estimating $\alpha$ might carry even more bound improvements.
>
> ____
> - **Q6**: Is it possible to use Clark Jacobians for ReLU NN verification with VPN?
>
> With what we know now, **our method cannot be applied to non-twice-differentiable functions like ReLU NNs**. $\alpha$ convexification requires the original function to be twice-differentiable. Clark Jacobians might be a possible workaround towards the estimation of an $\alpha$ that guarantees convexity, enabling the extension of $\alpha$-convexification and therefore our method for ReLU NNs. However, this needs to be studied carefully in the future.
>
> ____
> - **Q7**: Do lines 199-200 mean that VPN cannot be applied to NN verification?
>
> VPN can be applied to NN verification if the activation function is twice-differentiable and we are able to compute bounds of the Hessian matrix.  We have polished this sentence in the revised manuscript, see lines 199-200.
>
> ____
> We hope that our answers have addressed the questions of the reviewer; we would be happy to elaborate further if something remains unclear. Once again, we appreciate the thoughtful feedback of the reviewer.

---

> > ### Author Response · Authors · 2022-08-06
> > **Have the concerns of the reviewer been addressed?**
> >
> > Dear reviewer rPvn,
> >
> > We are thankful for your constructive feedback. We agree with the reviewer that the splitting strategy along with extension to twice-differentiable NNs, are interesting topics for future research on verification. However, our preliminary experiments indicate that the input splitting strategy has no relevant influence within our scenario.
> >
> > Since the discussion window is closing soon, we wanted to check whether the [newly added experiments](https://openreview.net/forum?id=gsdHDI-p6NI&noteId=HpWOZg6Px5M) and [clarifications](https://openreview.net/forum?id=gsdHDI-p6NI&noteId=gk2NKiyByOd_) address your questions. We are happy to answer any further inquiries on our framework the reviewer might have.

---

> > > ### Comment · Reviewer_rPvn · 2022-08-08
> > > **Post Rebuttal Comments**
> > >
> > > Dear Authors,
> > >
> > > Thanks for your excellent response which addressed all of my questions. I believe the ideas presented in this work are novel and have utility beyond polynomial networks. I am happy to see it accepted at NeurIPS and have increased my score.

---

> > > > ### Author Response · Authors · 2022-08-09
> > > > **Thank you for your response**
> > > >
> > > > Dear reviewer rPvn,
> > > >
> > > > We are grateful for your insightful comments and your acknowledgement of our updates. We also hope that our ideas will be useful to the verification community.

---

### Author Response · Authors · 2022-08-01
**General response: Improved experimental validation and method extension**

Thanks to the reviewers for their meaningful comments. The reviewers acknowledge the novelty of our method (reviewers rPvn and bVTt), the relevance Polynomial Network verification has (reviewers rPvn and sZV9), and the potential extension of our method towards verification of other twice-differentiable architectures (reviewers rPvn and sZV9). Motivated by your suggestions, we have improved our experimental validation in two fronts:

1) We include a comparison with the Transformer verification method DeepT-Fast (Bonaert et al., 2021) as requested by reviewer rPvn.

2) We extend our method to work with a different $\alpha$ parameter for each input dimension (Non-uniform diagonal shift) as requested by reviewer sZV9.

Finally, we analyze the possible improvements of our input set splitting strategy as suggested by reviewers rPvn and sZV9. We have revised the manuscript with changes highlighted in red for convenience.
## Comparison with DeepT-Fast.
As reviewer rPvn identified, any abstraction based verification method handling multiplications could be used for PN verification. We use DeepT-Fast to compute a lower bound of the global minima.

Firstly we analyze the average difference between the PGD upper bound and the lower bound provided by this method. In Fig. 3 of the manuscript (https://imgur.com/a/BJXbqjY), we observe that on the one hand, DeepT-Fast provides consistently tighter bounds than IBP in all the studied cases, proving the point of reviewer rPvN. But, on the other hand, DeepT-Fast provides much looser bounds than the ones obtained with our method.

Secondly, we test the performance when employing DeepT-Fast for lower bounding the global minima in our BaB algorithm.

|         |           |        |            | IBP     |        | DeepT-Fast |        | VPN            |        |         |
|:-------:|:---------:|:------:|:----------:|:-------:|:------:|:----------:|:------:|:--------------------------:|:------:|:------:|
| Dataset | Model     | Acc.\% | $\epsilon$ |         |        | (Bonaert et al., 2021)  |        |  |         |
|         |           |        |            | Time(s) | Ver.\% | Time(s)    | Ver.\% | Time(s)                    | Ver.\% | U.B. |
| MNIST   | PN\_Conv4 | 98.6   | 0.015      | 0.3     | 0.0    | 2.3        | 91.3   | 50                         | **96.3**   | 96.4 |
|         |           |        | 0.026      | 0.4     | 0.0    | 3.7        | 59.7   | 69                         | **92.9**   | 94.8 |
|         |           |        | 0.3        | 0.6     | 0.0    | 0.7        | 0.0    | 13.8                       | 0.0    | 0.0  |
| CIFAR10 | PN\_Conv2 | 63.5   | 1/255      | 0.3     | 0.0    | 2.0        | 23.3   | 136.2                      | **44.4**   | 44.6 |
|         |           |        | 2/255      | 0.5     | 0.0    | 0.6        | 1.4    | 89.2                       | **25.4**  | 27.5 |
|         | PN\_Conv4 | 62.6   | 1/255      | 0.4     | 0.0    | 2.2        | 19.5   | 274.6                      | **45.5**   | 46.7 |
|         |           |        | 2/255      | 0.5     | 0.0    | 0.5        | 0.5    | 224.1                      | **16.5**  | 30.5 |
| STL10   | PN\_Conv4 | 38.1   | 1/255      | 3.4     | 0.0    | 26.0       | 14.7   | 2481.0                     | **21.7**   | 21.9 |

Verification results from Tab. 2 are in line with results obtained in Fig. 3 regarding tightness of the lower bounds. With IBP we are not able to verify any image. With DeepT-Fast, the lower bounds obtained are tighter and we are able to effectively verify a large percentage of images, but the **percentage of verified images is not as high as when employing VPN**. This is logical as in Fig. 3 we observed $\alpha$-convexification bounds were much tighter than DeepT-Fast. This experiment has been added in Sec. 5 of the revised manuscript.

## Non-uniform diagonal shift.
Reviewer sZV9 avidly pointed out that the sumatory in Eq. 5 can be split. This is known as Non-uniform diagonal shift and is covered in (Adjiman and Floudas, 1998).
One might think that computing one $\alpha_{i}$ per input dimension is more expensive than computing a single $\alpha$ in the uniform diagonal shift setup. But, for computing the $\alpha_{i}$s there is **no need to perform a power method** as in the uniform diagonal shift setup. We observe:
- In the Non-uniform diagonal shift case, computing $\alpha_{i}$s is **faster and non-numerical**.
- However, lower bounds obtained in the verification of relevant PNs (degrees 2-4) are **looser than the Uniform diagonal shift variant**. See Fig. 3 (https://imgur.com/a/BJXbqjY).

### References
Bonaert et al., Fast and precise certification of transformers, PLDI 2021.

Adjiman and Floudas, A global optimization method, αbb, for general twice-differentiable constrained nlps — i. theoretical advances, Computers & Chemical Engineering 1998.

Botoeva et al., Efficient Verification of ReLU-Based Neural Networks via Dependency Analysis, AAAI 2020.

---

> ### Author Response · Authors · 2022-08-01
> **General response: On the input set splitting strategies.**
>
> We agree with the reviewers that the input splitting strategies are an important asset for verification. Despite being naive, the widest interval selection splitting strategy guarantees convergence of the BaB to the global minima. This has been highlighted in lines 133-134. We find the recommended strategies are not (easily) applicable as we elaborate below. Concretely:
> - The input splitting strategy proposed in (Botoeva et al., 2020) **cannot be applied to PN verification** due to the employed ReLU activation function in their splitting strategy.
> - The input splitting strategy proposed by reviewer sZV9 **needs to be proven to guarantee convergence** to the global minima. Also, **some aspects remain unclear** as in which point to evaluate the Hessian matrix to get its diagonal.
>
> We agree with reviewers rPvn and sZV9 in that a more optimal splitting strategy could be used. Branching is a key aspect of any Branch and Bound algorithm. Nevertheless, in our scenario, splitting strategies need to be chosen so that 1) convergence to the global minima is guaranteed and 2) the specific characteristics of PN verification are taken into account. We believe this is a promising research direction and thank reviewers rPvn and sZV9 for highlighting this aspect.
>
> ### References
> Botoeva et al., Efficient Verification of ReLU-Based Neural Networks via Dependency Analysis, AAAI 2020.

---

### Meta-Review · Area_Chair_qyS3 · 2022-08-26

**Recommendation:** Accept
**Confidence:** Certain

**Metareview:**

The authors propose a novel verification technique for polynomial neural networks. They compare their approach against competitive baselines and demonstrates improvements in the quality of bounds obtained and their ability to verify input-output properties.

The reviewers agreed that the paper contains novel and interesting ideas and all concerns brought up by the reviewers were thoroughly addressed in the rebuttal phase.

Hence, I recommend acceptance.

**Award:**

No

---

### Decision · Program_Chairs · 2022-09-14

Accept